# L-Serine Supplementation Blunts Fasting-Induced Weight Regain by Increasing Brown Fat Thermogenesis

**DOI:** 10.3390/nu14091922

**Published:** 2022-05-04

**Authors:** Elena López-Gonzales, Lisa Lehmann, Francisco Javier Ruiz-Ojeda, René Hernández-Bautista, Irem Altun, Yasuhiro Onogi, Ahmed Elagamy Khalil, Xue Liu, Andreas Israel, Siegfried Ussar

**Affiliations:** 1RG Adipocytes & Metabolism, Institute for Diabetes & Obesity, Helmholtz Center Munich, 85764 Munich, Germany; elena.lopez@helmholtz-muenchen.de (E.L.-G.); lisa@suwandhi.de (L.L.); fjrojeda@gmail.com (F.J.R.-O.); renejavier.hernandez11@gmail.com (R.H.-B.); irem.altun@helmholtz-muenchen.de (I.A.); yasuhiro.onogi@helmholtz-muenchen.de (Y.O.); ahmed.khalil@helmholtz-muenchen.de (A.E.K.); xue.liu@helmholtz-muenchen.de (X.L.); andreas.israel@helmholtz-muenchen.de (A.I.); 2German Center for Diabetes Research (DZD), 85764 Munich, Germany; 3Department of Biochemistry and Molecular Biology II, Faculty of Pharmacy, University of Granada, 18071 Granada, Spain; 4Department of Medicine, Technical University of Munich, 80333 Munich, Germany

**Keywords:** L-serine, BAT, fasting, obesity, body weight, weight regain, energy expenditure

## Abstract

Weight regain after fasting, often exceeding the pre-fasting weight, is a common phenomenon and big problem for the treatment of obesity. Thus, novel interventions maintaining reduced body weight are critically important to prevent metabolic disease. Here we investigate the metabolic effects of dietary L-serine supplementation, known to modulate various organ functions. C57BL/6N-Rj male mice were supplemented with or without 1% L-serine in their drinking water and fed with a chow or high-fat diet. Mice were fed either *ad libitum* or subjected to repeated overnight fasting. Body weight, body composition, glucose tolerance and energy metabolism were assessed. This was combined with a detailed analysis of the liver and adipose tissues, including the use of primary brown adipocytes to study mitochondrial respiration and protein expression. We find that L-serine supplementation has little impact on systemic metabolism in *ad libitum*-fed mice. Conversely, L-serine supplementation blunted fasting-induced body weight regain, especially in diet-induced obese mice. This reduction in body weight regain is likely due to the increased energy expenditure, based on elevated brown adipose tissue activity. Thus, L-serine supplementation during and after weight-loss could reduce weight regain and thereby help tackle one of the major problems of current obesity therapies.

## 1. Introduction

Obesity has reached endemic levels worldwide. In addition to the direct effects of excessive body fat, obesity increases the risk for the development of type 2 diabetes, cardiovascular disease and certain types of cancer [1,2,3]. Current pharmacological treatments show only moderate efficacy in weight loss and surgical interventions are not available to most patients [4]. Moreover, lifestyle interventions, such as changes in diet and exercise, do not provide long-lasting weight reduction and result in weight regain, often exceeding initial body weight [5]. Weight regain is especially problematic as it increases the risk for metabolic disease due to an inflammatory memory of previous obesity compared to subjects with the same body fat content without prior weight loss [6,7,8]. Thus, novel approaches promoting sustained weight loss or preventing weight regain are urgently needed. Importantly, the ideal intervention would be initiated before the onset of metabolic complications, thus requiring a very high safety profile. In this context, dietary supplements could support other treatment strategies. Among the various supplements, L-serine has shown interesting beneficial effects on insulin secretion and the protection of pancreatic beta cells in models of type 1 diabetes as well as in reducing food intake [9,10]. Furthermore, human data show an association of high L-serine concentrations with improved insulin secretion and sensitivity as well as better glucose tolerance [11].

L-Serine is a central metabolite for various cellular functions. It is an important proteinogenic amino acid. Moreover, serine is a substrate for purine, phosphatidylserine and sphingolipids synthesis and NADPH [12,13,14]. Thus, L-serine supplementation could promote tumor growth [15,16]. However, L-serine supplementation is generally considered safe by the Food and Drug Administration (FDA) [17]. Moreover, L-serine supplementation shows neuroprotective effects and the enantiomer of L-serine, D-serine, is an important neurotransmitter functioning as a co-agonist of the N-methyl-D-aspartate (NMDA) receptor [3] and deficiency of D-serine is associated with the development of mental disorders such as schizophrenia [18]. L-serine supplementation also ameliorates DSS (dextran sodium sulfate)-induced colitis and gut microbiota composition in mice [19]. Thus, various health-promoting effects of L-serine supplementation have been documented. However, surprisingly little is known on the role of L-serine in the regulation of body weight, despite a role of D-serine in reducing high-fat diet consumption and body weight gain [18] and a weight-reducing effect of long term L-serine supplementation [9]. Here, we describe the metabolic effects of L-serine supplementation during *ad libitum* access to food as well as upon fasting and refeeding. 

## 2. Materials and Methods

### 2.1. Animals

Male mice aged 3 or 7 weeks old C57BL/6N-Rj were purchased from Janvier (France), kept under a cycle of 12 h light and 12 h dark at a temperature of 22 °C and fed with either regular chow diet (CD) (Altromin 131, Lage, Germany) or a 58% high-fat diet (HFD, Research Diets D12331, Research Diets, Inc. New Brunswick, NJ, USA) *ad libitum* for 8 or 16 weeks. Mice were provided with sterile water or sterile water supplemented with 1% L-serine (84959-100G; Sigma-Aldrich, Schnelldorf, Germany). Body weight and blood glucose were monitored weekly. We assigned 6–8 mice per condition. Body composition was analyzed by nuclear magnetic resonance (EchoMRI, Houston, TX, USA) at the beginning and the end of the experiment. A pyruvate tolerance test (PTT) was performed after overnight fasting (16 h) by intraperitoneally injecting 2 g/kg sodium pyruvate (Sigma-Aldrich, Taufkirchen, Germany). A glucose tolerance test (GTT) was performed after 4 h fasting by injecting 2 g/kg glucose (Braun, Melsungen, Germany). Energy expenditure (EE), respiratory exchange ratio (RER) and food and water intake were measured by indirect calorimetry (TSE PhenoMaster, Bad Homburg, Germany). Random fed or overnight-fasted mice were euthanized by overdosing anesthetics after 8 or 16 weeks. Animal experiments were performed according to the German animal welfare law, guidelines and regulations of the district of Upper Bavaria (Munich, Germany), protocol number 55.2-1-54-2532-52-2016.

### 2.2. RNA Isolation, Transcription and RT-PCR

Total RNA was extracted from frozen perigonadal fat (PGF), subcutaneous fat (SCF), BAT (brown adipose tissue) and the hypothalamus using Qiazol lysis reagent (Qiagen, Hilden, Germany) and the RNeasy Mini kit (Qiagen, Hilden, Germany). RNA was reverse transcribed to cDNA (High-Capacity cDNA Reverse Transcription Kit; Applied Biosystems, Thermo Fisher Scientific, Darmstadt, Germany) following the manufacturer’s instructions. Quantitative PCR was performed using gene-specific primers (Appendix A) and the iTaq Universal SYBR Green Supermix (Biorad, Feldkirchen, Germany) in a CFX384 Touch (Biorad, Feldkirchen, Germany). Gene expression was analyzed with the BioRad CFX Manager 3.1 Software version 3.1.1517.0823 (Biorad, Feldkirchen, Germany) and normalized by TATA box-binding protein (Tbp). Expression levels were calculated with the ΔCt method.

### 2.3. Western Blot for Protein Expression Measurement

Snap frozen tissues or cells were lysed with RIPA buffer (50 mM Tris; 150 mM sodium chloride; 1 mM Ethylenediaminetetraacetic acid (EDTA); 1% Triton-X100; and 0.1% sodium dodecyl sulfate (SDS), pH 7.4) supplemented with 1% protease inhibitor cocktail (Sigma-Aldrich, Taufkirchen, Germany), 1% phosphatase inhibitor II (Sigma-Aldrich, Taufkirchen, Germany), 1% phosphatase inhibitor III (Sigma-Aldrich, Taufkirchen, Germany). Protein concentration was measured with the BCA protein assay kit (Thermo Fisher Scientific, Darmstadt, Germany). Protein extracts were denatured by boiling them at 70 °C for 10 min with 2.5% β-mercaptoethanol (Carl Roth, Karlsruhe, Germany) in LDS sample buffer (NuPAGE LDS, Invitrogen, Thermo Fisher Scientific, Darmstadt, Germany) and loaded on 10% SDS poly-acrylamide gels. Proteins were blotted onto 0.45 µm polyvinylidene difluoride (PVDF) membranes (Thermo Fisher Scientific, Darmstadt, Germany) and blocked at room temperature for 1 h with 5% skim milk (Biomol, Hamburg, Germany) in 0.1% Tween 20 containing TBS (TBS-T). Then, membranes were incubated at 4 °C overnight with the corresponding primary antibodies diluted in 5% bovine serum albumin (BSA) (Carl Roth, Karlsruhe, Germany) in TBS-T following incubation with the corresponding HRP-conjugated secondary antibodies in 5% skim milk in TBS-T. Primary antibodies: β-actin HRP (Santa Cruz Biotechnology, Inc., Dallas, TX, USA; 1:1000), Hormone-Sensitive Lipase (HSL) (Cell Signalling, Beverly, MA, USA; 1:1000), phospho-HSL Ser563 (Cell Signaling, Beverly, MA, USA; 1:1000), phospho-HSL Ser565 (Cell Signaling, Beverly, MA, USA; 1:1000), phosphor-HSL Ser660 (Thermo Fisher Scientific, Darmstadt, Germany; 1:1000) and Uncoupled protein 1 (UCP1) (D9D6X. Cell Signaling, Beverly, MA, USA; 1:1000). Secondary antibody: goat α-rabbit HRP (Invitrogen, Thermo Fisher Scientific, Darmstadt, Germany; 1:5000). Membranes were developed with HRP substrate (immobilon western chemilum HRP substrate; EMD Millipore Corporation, Burlington, MA, USA) and imaged with a Chemidoc MP (BioRad, Feldkirchen, Germany). Protein quantification was performed using ImageJ version 1.52v software.

### 2.4. Histology

Tissues were fixed in 4% paraformaldehyde/phosphate-buffered saline (PFA/PBS) overnight and stored in 70% ethanol. Tissues were dehydrated with ethanol, cleared with xylol and incubated in paraffin overnight. The next day, tissues were paraffin-embedded in cassettes, cut into 2 µm sections with the semi-automated microtome HM390E (Thermo Fisher Scientific, Waldorf, Germany) and stained with hematoxylin and eosin (H&E). Images were taken with a Zeiss Scope A1 (Carl Zeiss Canada, Toronto, ON, Canada).

### 2.5. Serum and Liver Hormones and Metabolites

Serum insulin levels were determined with the Mouse Ultrasensitive Insulin ELISA kit (Alpco, Salem, NH, USA). Glycogen content in the liver was measured either with a starch assay (r-Biopharm, Pfungstadt, Germany) or a Glycogen colorimetric/fluorimetric assay kit (Biovision, Milpitas, CA, USA). Triglyceride content was measured with the Triglyceride Quantification colorimetric/fluorimetric kit (Biovision Milpitas, CA, USA). L-serine was quantified in tissues with the DL-Serine Assay Kit (Fluorometric) (Biovision, Milpitas, CA, USA). 

### 2.6. Ex Vivo Lipolysis Assay

PGF pads were obtained from wild-type mice after sacrifice by cervical dislocation. Tissues were cut into small pieces (3–5 mg) and pre-equilibrated in pre-warmed Dulbecco’s Modified Eagle Medium (DMEM) (31966047; Life Technologies Thermo Fisher Scientific, Darmstadt, Germany) at 37 °C, 5% CO_2_ for 30 min. Then, tissues were incubated with 200 µM L-serine, 400 µM L-serine, with or without 1 µM isoproterenol (Sigma Aldrich, Taufkirchen, Germany) or with 1 µM isoproterenol alone in DMEM for 3 h at 37 °C, 5% CO_2_. After the incubation, the media were used for glycerol quantification with the free glycerol colorimetric assay kit (Biovision, Milpitas, CA, USA) and fat pads were delipidated with 2:1 chloroform/methanol solution at 37 °C with shaking for 1 h. Delipidated tissues were transferred to a 0.3 N NaOH/0.1% SDS solution and kept at 40 °C overnight with shaking for protein extraction. Protein measurement was done with the BCA protein assay kit (Thermo Fisher Scientific, Darmstadt, Germany). Lipolysis was calculated based on free glycerol content in the assay media normalized by protein content.

### 2.7. Primary Brown Adipocytes Culture

Pre-adipocytes were obtained from the stromal vascular fraction (SVF) of the interscapular BAT of wild-type mice. BAT was minced into small pieces and digested using media containing 1% BSA (Carl Roth, Karlsruhe, Germany) and 1 mg/mL Collagenase type IV (Gibco, Thermo Fisher Scientific, Darmstadt, Germany) in DMEM (31966047; Life Technologies, Thermo Fisher Scientific, Darmstadt, Germany), for 45 min at 37 °C with shaking. After washing and centrifugation twice for 5 min at 500× *g*, SVF was re-suspended in DMEM supplemented with 10% fetal bovine serum (FBS) (Gibco, Thermo Fisher Scientific, Darmstadt, Germany), 1% penicillin/streptomycin (Gibco, Thermo Fisher Scientific, Darmstadt, Germany) and 100 µg/mL Normocin (InvivoGen, Toulouse France). Preadipocytes were grown in a 6-well plate and re-plated in either 6-well or 24-well plates. The cell culture medium was changed every other day. At confluence, preadipocytes were induced for 2 days with induction media (DMEM supplemented with 10% FBS, 0.5 mM 3-Isobutyl-1-methylxanthine (IBMX) (Sigma Aldrich, Taufkirchen, Germany), 5 µM dexamethasone (Sigma Aldrich, Taufkirchen, Germany), 125 µM indomethacin (Sigma Aldrich, Taufkirchen, Germany), 100 nM human insulin (Sigma Aldrich, Taufkirchen, Germany) and 1 nM 3,3’,5’-Triiodo-L-thyronine (T3) (Calbiochem, Sigma Aldrich, Taufkirchen, Germany). Cells were then incubated with differentiation media (DMEM, 10% FBS with 100 nM insulin and 1 nM T3) until the day of the experiment.

### 2.8. Cellular Respiration

Primary brown preadipocytes were cultured and differentiated on XF24 Seahorse plates (Agilent technologies Inc., Santa Clara, CA, USA). For each biological replicate, brown preadipocytes from two mice were pooled. At day 7 of differentiation, brown adipocytes were pre-treated with 400 µM L-serine for 3 h. Prior to measurement and following the pre-treatment, DMEM was changed to XF Assay Medium-Modified DMEM Agilent technologies Inc., Santa Clara, CA, USA) supplemented with or without 400 µM L-serine. Assay ports were loaded with: (A) assay media, 4 mM L-serine, 10 µM isoproterenol or 4 mM L-serine and 10 µM isoproterenol, (B) 20 µg/mL oligomycin (Merck, Darmstadt, Germany), (C) 20 µM carbonyl cyanide-p-trifluoromethoxyphenylhydrazone (FCCP) (R&D systems, Wiesbaden, Germany), (D) 25 µM rotenone; 25 µM antimycin A (Sigma Aldrich, Taufkirchen, Germany) and 1 M 2-deoxy-D-glucose (Alfa Aesar, Kandel, Germany). Oxygen consumption rates (OCR) were measured at day 7 of differentiation with the Seahorse XF24 Analyzer (Seahorse Bioscience-Agilent, North Billerica, MA, USA). Each cycle consisted of 2 min mixing, 2 min waiting and 2 min measuring. Data analysed with Wave software version 2.6.1.53 (Agilent technologies Inc., Santa Clara, CA, USA).

### 2.9. Adipocyte Size Measurements

Adipocyte size distribution as shown in Appendix A was measured as described in [18]. Adipocyte sizes shown in Appendix A were measured using the Adiposoft plug-in for ImageJ version 1.52v (https://imagej.net/plugins/adiposoft; Accessed on 21 April 2022 using 3 pictures taken per adipose tissue from each mouse. Outliers were removed using ROUT (Q = 1%) in GraphPad Prism 8 version 8.4.3 (GraphPad software, La Jolla, CA, USA). 

### 2.10. Statistical Analysis

All statistics were calculated using GraphPad Prism 8 version 8.4.3 (GraphPad software, La Jolla, CA, USA). Data are presented as mean ± standard error of the mean (SEM) unless stated differently in the figure legend. Statistical significance was determined by unpaired Student’s *t*-test between two groups or, for multiple comparisons, using One- or Two-Way ANOVA, followed by Tukey’s multiple comparison test, or as stated in the respective figure legend. A univariate general linear model was performed to study the effect of the diet on energy expenditure, adjusting for body weight in SPSS version 24 (SPPS Inc., Chicago, IL, USA). The mean difference was calculated for each paired comparison. Differences reached statistical significance with *p* < 0.05.

## 3. Results

### 3.1. L-Serine Supplementation Does Not Alter Body Weight or Glucose Homeostasis in Ad Libitum-Fed Young Mice

To test the effects of L-serine supplementation on body weight development and metabolic function, 4-week-old mice were supplemented with 1% L-serine in their drinking water for 8 weeks and fed either a CD or a HFD. L-serine supplementation did not alter body weight (Figure 1a) or body composition (Figure 1b). Similarly, we did not observe changes in body weight when 8-week-old mice, fed either a CD or HFD were supplemented with L-serine for 16 weeks (Appendix A). L-serine supplementation also did not alter parameters of glucose homeostasis, such as randomly-fed blood glucose (Figure 1c) and insulin levels (Figure 1d) or glucose tolerance (Figure 1e). H&E stainings of the liver, subcutaneous (SCF) and perigonadal (PGF) fat as well as qPCR analysis in adipose depots did not indicate changes in inflammation (Figure 1f and Appendix A) or adipocyte cell size (Appendix A) between the control and L-serine-supplemented mice of each diet group. However, HFD feeding increased the proportion of large adipocytes compared to CD-fed mice, irrespective of L-serine supplementation. L-serine supplementation did not affect hepatic triglyceride (Appendix A) or serine levels (Appendix A). However, we observed a significant decrease in the liver weight of serine-supplemented mice upon HFD feeding but not CD feeding (Appendix A), as well as a trend towards reduced glycogen levels in both CD-fed and HFD-fed mice (Appendix A). The expression of Phosphoenolpyruvate carboxykinase (Pepck) was increased in the CD-fed mice supplemented with L-serine, which was not observed upon HFD feeding (Appendix A). We did not observe any differences in pyruvate tolerance between the serine-supplemented and control animals on either diet after 1 week of supplementation (Appendix A). 

### 3.2. L-Serine Supplementation Blunts Body Weight Regain after Repeated Overnight Fasting

Pyruvate tolerance tests (PTTs), in contrast to insulin and glucose tolerance tests (GTTs), require an overnight fast to deplete hepatic glycogen stores. This results in a ~10–15% weight loss in mice, which is usually regained within a week. We observed a reduced weight regain in L-serine-supplemented HFD-fed mice following the PTT (Figure 2a). To test the role of L-serine supplementation in weight regain, we subjected CD and HFD mice with or without L-serine supplementation to additional overnight fasting. Indeed, L-serine supplementation in HFD-fed mice reduced body weight regain after each overnight fast (Figure 2a). The weight difference between the control and L-serine-supplemented mice increased with every fasting/refeeding cycle (Figure 2a). Liver weight did not change in any condition (Appendix A), but triglyceride content decreased in the CD with L-serine supplementation (Appendix A). In line with the triglyceride levels, liver histology showed fewer lipids in the CD-fed mice upon L-serine supplementation. Interestingly, L-serine seemed to decrease the size of the lipid droplets in the HFD, but not the number (Appendix A). An assessment of body composition showed that the decrease in body weight was due to a reduction in fat but not lean mass (Figure 2b). This was also seen in a reduction in the weight of individual fat depots (Figure 2c) and the size of subcutaneous and perigonadal adipocytes (Figure 2d and Appendix A). A gene expression analysis of TNFα IL-6 and CD68 expression in SCF and PGF revealed a significant downregulation of TNFα and a strong trend toward reduced CD68 expression in the PGF of HFD-fed L-serine-supplemented mice compared to HFD-fed controls (Appendix A). We did not observe any differences in glucose tolerance between groups as the repeated fasting retained normal glucose tolerance in control HFD-fed mice when compared to the CD-fed mice (Figure 2e). 

### 3.3. L-Serine Supplementation Increases Energy Expenditure following an Overnight Fast

To study the metabolic changes induced by the interaction of fasting and L-serine supplementation, we studied CD-fed and HFD-fed mice either with or without serine supplementation in metabolic cages (Appendix A). After acclimatization, the CD (after 7 weeks) and HFD (after 8 weeks) mice were fasted for 16 h in metabolic cages and food intake, energy expenditure, activity and respiratory exchange ratios were assessed in the 72 h thereafter (Figure 3a). As expected, daily food intake and cumulative food intake were increased in the CD-fed mice following an overnight fast compared to *ad libitum*-fed mice (Figure 3b and Appendix A). A similar trend was observed in the HFD-fed mice. The increase in daily food intake was largely attributed to increased feeding during the light cycle. L-serine supplementation increased food intake in CD-fed, but not HFD-fed mice (Figure 3b and Appendix A). A similar trend was observed for water intake (Appendix A). In line with the changes in food intake, we observed increased Neuropeptide Y (Npy) and decreased Proopiomelanocortin (Pomc) expression in the hypothalamus of overnight-fasted mice fed a CD, whereas no statistically significant differences were observed in mice fed a HFD at the end of the 16-week L-serine supplementation (Appendix A). No differences in Npy or Pomc expression between the control and L-serine-supplemented mice were observed in either experimental condition (Appendix A). Thus, L-serine supplementation increased chow diet intake following an overnight fast compared to control animals. Conversely, there was a trend for slightly reduced food intake during the light phase in the HFD-fed L-serine-supplemented mice following an overnight fast. These data suggested that differences in food intake per se could not explain the reduced weight regain of HFD-fed mice upon serine supplementation. To this end, we investigated whether energy expenditure and substrate utilization, assessed by the respiratory exchange ratio (RER), were altered by serine supplementation. *Ad libitum* CD-fed mice showed the typical diurnal rhythm of substrate utilization with the highest RER during the dark phase (active phase) and the lowest values, indicating fatty acid oxidation, during the light phase (resting phase), with no differences between the control and L-serine-supplemented mice (Figure 3c). Fasting reduced the RER in all groups and diets (Appendix A). This also demonstrated that consumption of L-serine through the drinking water did not significantly contribute to energy expenditure, as this would raise the RER. Refeeding rapidly increased the RER, which remained elevated throughout the remaining time measured (Appendix A). The absence of diurnal fluctuations can be explained by the increased food intake during the light phase. The RER of control mice increased its amplitude during the 72 h following fasting, whereas the L-serine-supplemented mice remained at close to one throughout the whole time. This could result from the increased food intake observed in the L-serine-supplemented mice or indicate preferential oxidation of carbohydrates. The RER pattern was similar in the HFD-fed mice, albeit the overall amplitude of oscillation was reduced due to the increased dietary fats (Figure 3c). Moreover, fasted mice returned more rapidly after the next dark cycle to the diurnal oscillation observed in *ad libitum*-fed mice (Appendix A). Interestingly, L-serine supplementation lowered the RER in the HFD-fed mice compared to controls, potentially indicating increased adipose tissue lipolysis (Appendix A). To this end, we quantified HSL phosphorylation at the activating sites Ser563 and Ser660, as well as the inhibitory Ser565 site in SCF and PGF following an overnight fast (Figure 3d and Appendix A). We observed increased Ser660 and reduced Ser565 phosphorylation in HSL in the HFD compared to the CD-fed animals in PGF. However, we did not observe any differences between the control and L-serine-supplemented mice irrespective of the diet. In addition, serine supplementation led to a reduction in phosphorylation of Ser565 in the CD compared to the HFD in SCF (Figure 3d). Thus, these data did not suggest an increased lipolysis in white adipose tissue in the interaction of fasting and L-serine supplementation. Moreover, ex vivo stimulation of PGF with either 200 µM or 400 µM L-serine did not increase lipolysis as measured by glycerol release into the medium (Figure 3e). We also did not observe an effect of 400 µM L-serine on isoproterenol-induced lipolysis (Figure 3e). 

### 3.4. L-Serine Supplementation Activates Brown Adipose Tissue Thermogenesis after Fasting

Neither food intake nor changes in lipolysis explained the decreased weight regain of L-serine-supplemented mice upon HFD feeding. In *ad libitum* CD-fed mice, L-serine supplementation decreased energy expenditure (EE) (*p* = 0.002) (Figure 4a). Conversely, EE was increased with L-serine supplementation following an overnight fast (*p* = 0.0003), which is in line with the food intake data (Figure 3b). Similarly, L-serine supplementation showed a trend towards reduced EE in *ad libitum* HFD-fed mice versus an increase following an overnight fast (Figure 4a). The inverse relation between body weight and EE in fasted control animals complicates the interpretation of the latter results. Nevertheless, these data suggest that increased EE could explain the observed reduction in weight regain in L-serine-supplemented HFD-fed mice. The total activity was not changed between groups (Appendix A). Thus, we investigated a potential role of brown adipose tissue (BAT) activity in the observed phenotype. Histological analysis of BAT showed that upon fasting, L-serine supplementation reduced lipid droplet size in both the CD-fed and HFD-fed mice (Figure 4b), which was also reflected by decreased BAT weight in CD-fed mice and a trend in HFD-fed mice (Appendix A). Gene expression of important thermogenesis-related genes were not changed in the BAT of any experimental group or any diet (Appendix A). However, uncoupled protein 1 (UCP1) protein levels were significantly higher in L-serine-supplemented fasted mice fed a CD. This was not observed in HFD-fed mice, which generally had much lower UCP1 protein levels (Figure 4c and Appendix A). Thus, we tested if L-serine has direct effects on mitochondrial respiration in primary brown adipocytes. Pretreatment with L-serine or an acute injection did not alter basal respiration. However, L-serine potentiated isoproterenol-induced and maximal respiration (Figure 4d). 

## 4. Discussion

Here we performed a detailed characterization of the metabolic effects of L-serine supplementation to the drinking water of mice fed *ad libitum* or subjected to repeated overnight fasting. We show that L-serine supplementation had very little metabolic effect in mice fed either a CD or HFD *ad libitum*. We found a reduction in liver weight upon HFD feeding, which was not associated with changes in hepatic triglyceride levels, but in reduced glycogen content. The reduction in liver glycogen was also observed in CD-fed mice and could indicate a role of L-serine in regulating hepatic glucose storage or metabolism. In addition, we observed a decrease in cumulative CD, but not HFD, intake upon L-serine supplementation. This had no effects on body weight, indicating that either small metabolic adjustments in energy expenditure or the caloric content of consumed L-serine compensate for the reduced food intake. Nevertheless, neither of these changes in liver glycogen metabolism or food intake altered systemic glucose tolerance. 

In contrast to this, a combination of L-serine supplementation and overnight fasting had profound metabolic effects. Most importantly, L-serine supplementation blunted weight regain upon starvation in HFD-fed mice. The difference in body weight between the control and L-serine-supplemented HFD-fed mice increased with every fasting intervention. The differences in body mass were due to differences in fat and not lean mass. We observed a similar trend in CD-fed mice, although the differences did not reach statistical significance. Nevertheless, these data indicate that there is a diet-independent interaction between fasting and L-serine supplementation in regulating weight regain. Mechanistically, our data show that L-serine supplementation results in increased energy expenditure following an overnight fast. Curiously, the effects were stronger in CD-fed than HFD-fed mice, albeit the overall effect on body weight regain was larger in HFD-fed mice. This is most likely due to an increase in the food intake of fasted L-serine-supplemented mice receiving a CD, compensating for the increased energy expenditure. Moreover, the CD-fed L-serine-supplemented mice showed a higher RER following an overnight fast, which was not observed in the HFD fasted mice, indicating differences in substrate utilization and availability of the different diets upon recovery from the overnight fast. Further analysis revealed increased BAT activity as a potential mechanism underlying the decreased weight regain of L-serine-supplemented mice following an overnight fast. Indeed, experiments with in vitro differentiated primary brown adipocytes confirmed the ability of L-serine to increase oxygen consumption and maximal respiration. However, it remains to be determined how L-serine molecularly interacts with the fasting response and what the exact molecular effectors are of this fasting response. Identifying specific factors is complex due to the profound effects of fasting on systemic metabolism and body weight regulation in mice [20,21,22]. However, prolonged fasting for 5–6 weeks in obese subjects increased plasma L-serine levels, which was detectable already after the first day of fasting [23]. Similar results were also observed in lean subjects [24]. The source of this circulating L-serine is unclear, but it could originate from skeletal muscle autophagy. Various other amino acids also play an important role during fasting, such as alanine, in the alanine cycle or in the regulation of body weight such as arginine, leucine, isoleucine and threonine. Supplementation with L-arginine reduced body weight and fat mass in rats [25,26] and humans [27]. Similarly, leucine, isoleucine [28] and threonine [29] supplementation decreased body weight in HFD-fed mice through mechanisms involving AMPKα activation [25] and browning in WAT [28]. It will be interesting to study if L-serine supplementation affects the blood levels of other amino acids, which could contribute to the observed phenotype. Nevertheless, an increase in circulating L-serine levels could be an endocrine signal during starvation indicating sufficient nutrient supply, resulting in an increase in activation of BAT and energy expenditure.

Most importantly, L-serine supplementation is generally recognized as safe (GRAS) by the FDA [17] and beneficial effects of L-serine supplementation have been described in animal models and humans. L-serine supplementation improved glucose homeostasis in non-obese diabetic (NOD) mice [30,31] and plasma L-serine levels were associated with improved insulin sensitivity in humans [11]. Moreover, experimental and clinical data suggest that L-serine supplementation improves hereditary sensory and autonomic neuropathy type 1 (HSAN1) disease [32,33]. Furthermore, ongoing clinical trials investigate L-serine supplementation for treatment in Alzheimer’s disease [34] and GRIN-related encephalopathy in children [35]. Thus, the potential effects of L-serine supplementation on body weight regain in humans could be easily studied. This will be greatly important as major species differences exist between mice and humans. Overnight fasting has much more pronounced metabolic consequences in mice than in humans due to much higher metabolic rate [36]. Thus, it will be important to define which hypocaloric weight loss program corresponds best to the overnight fasting described here. Moreover, we should bear in mind that BAT has a more important role in the regulation of systemic metabolism in mice than in humans. However, our data strongly suggest that L-serine supplementation during and after weight-loss interventions could significantly reduce weight regain and thereby help tackle one of the major problems of current obesity therapies.

## Figures and Tables

**Figure 1 nutrients-14-01922-f001:**
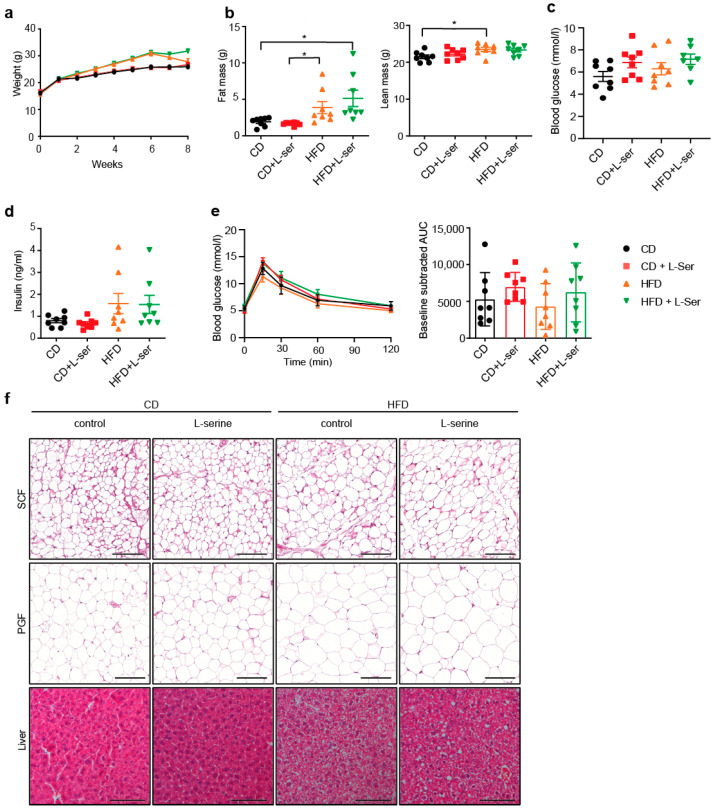
L-serine supplementation has no major effects on metabolism in *ad libitum*-fed mice. (**a**) Body weight of chow diet (CD) and high-fat diet (HFD) fed mice supplemented with or without L-serine for 8 weeks, starting at 4 weeks of age (*n* = 8). (**b**) Body composition (fat and lean mass) of CD-fed and HFD-fed mice with and without L-serine supplementation assessed after 8 weeks (*n* = 8). (**c**) Randomly fed blood glucose levels of CD-fed and HFD-fed mice with and without L-serine supplementation after 3 weeks (*n*= 8). (**d**) Randomly fed insulin levels of CD-fed and HFD-fed mice with and without L-serine supplementation after 3 weeks. (**e**) Glucose Tolerance Test (GTT) and baseline subtracted area under the curve (AUC) of CD -fed and HFD-fed mice with and without L-serine supplementation at week 6 (*n* = 8). (**f**) Representative H&E stainings of perigonadal fat (PGF), SCF (subcutaneous fat) and liver of random CD-fed and HFD-fed mice with or without L-serine supplementation after 8 weeks. Scale bars represent 100 µM. Data are shown as mean ± SEM, one or two-way ANOVA with Tukey’s post-test. * *p* < 0.05.

**Figure 2 nutrients-14-01922-f002:**
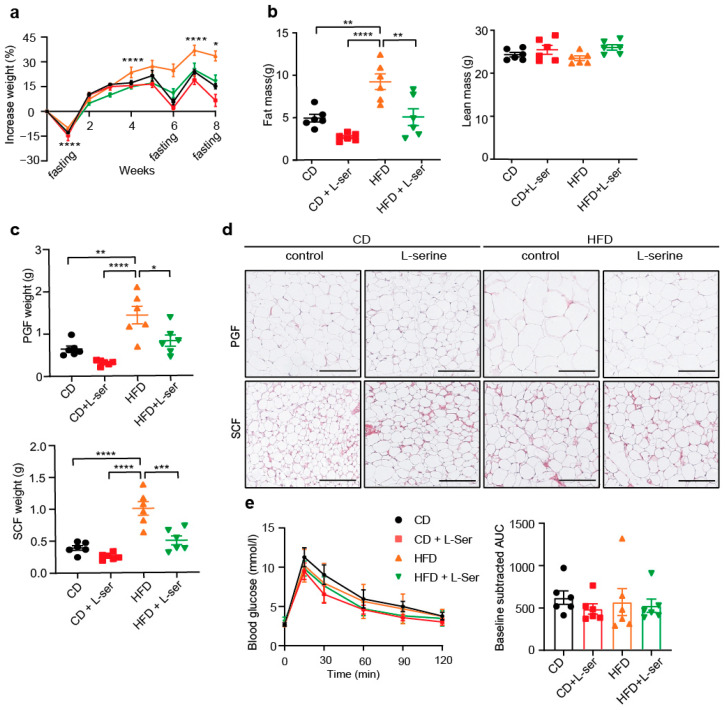
L-serine supplementation blunts fasting-induced body weight regain. (**a**) Body weight increase (%) of mice a fed chow diet (CD) or high-fat diet (HFD) with or without L-serine supplementation and repeated overnight fasting for 8 weeks starting at 8 weeks of age (*n* = 6). (**b**) Body composition (fat and lean mass) of mice shown in (**a**) after 8 weeks (*n* = 6). (**c**) Weight of perigonadal fat (PGF) and subcutaneous fat (SCF) of mice in (**a**) after week 8 (*n* = 6). (**d**) Representative H&E stainings of PGF and SCF from mice shown in (a) after 8 weeks. Scale bars show 100 µM. (**e**) Glucose Tolerance Test (GTT) of CD-fed and HFD-fed mice with and without L-serine supplementation at week 6 (*n* = 6). Data are shown as mean ± SEM, one or two-way ANOVA with Tukey’s post-hoc test. * *p* < 0.05, ** *p* < 0.01, *** *p* < 0.001, **** *p* < 0.0001.

**Figure 3 nutrients-14-01922-f003:**
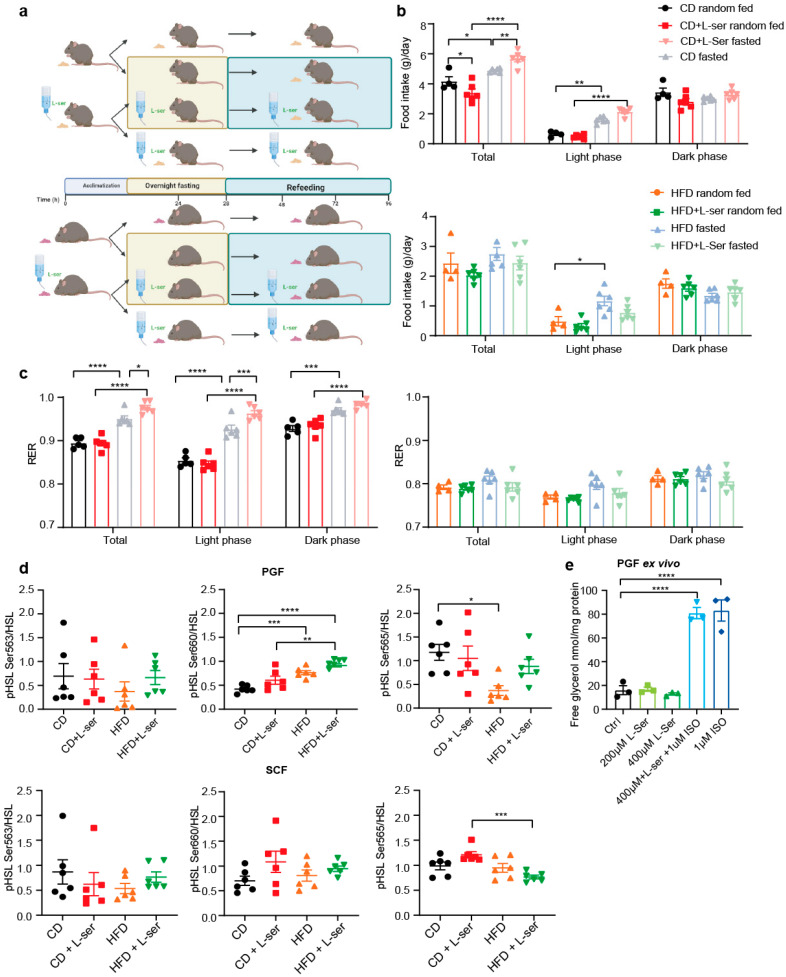
L-serine supplementation alters food intake and respiratory exchange ratio (RER), but not white adipose tissue (WAT) lipolysis. (**a**) Experimental setup for indirect calorimetry measurements. The figure was generated using Biorender.com. (**b**) Average food intake of random fed and fasted chow diet (CD) and high-fat diet (HFD)-fed mice with or without L-serine supplementation at week 7 (CD) or 8 (HFD) (CD random fed: *n* = 5; CD fasted, CD+L-ser random fed, CD+L-ser fasted: *n* = 6; HFD random fed: *n* = 4; HFD fasted, HFD+L-ser random fed, HFD+L-ser fasted: (*n* = 6). (**c**) Mean respiratory exchange ratio (RER) of random fed and fasted CD and HFD fed mice with or without L-serine supplementation at weeks 7 (CD) or 8 (HFD) (CD random fed: *n* = 5; CD fasted, CD+L-ser random fed, CD+L-ser fasted: *n* = 6; HFD random fed: *n* = 4; HFD fasted, HFD+L-ser random fed, HFD+L-ser fasted: *n* = 6). (**d**) Quantifications of Western blots for phosphorylated HSL (Ser563, Ser660, Ser565) and total HSL of PGF and SCF (*n* = 6) of fasted CD-fed and HFD-fed mice with or without L-serine supplementation. (**e**) Ex vivo lipolysis assay of PGF (*n* = 3). Data are shown as mean ± SEM, one or two-way ANOVA with Tukey’s post-hoc test. * *p* < 0.05, ** *p* < 0.01, *** *p* < 0.001, **** *p* < 0.001.

**Figure 4 nutrients-14-01922-f004:**
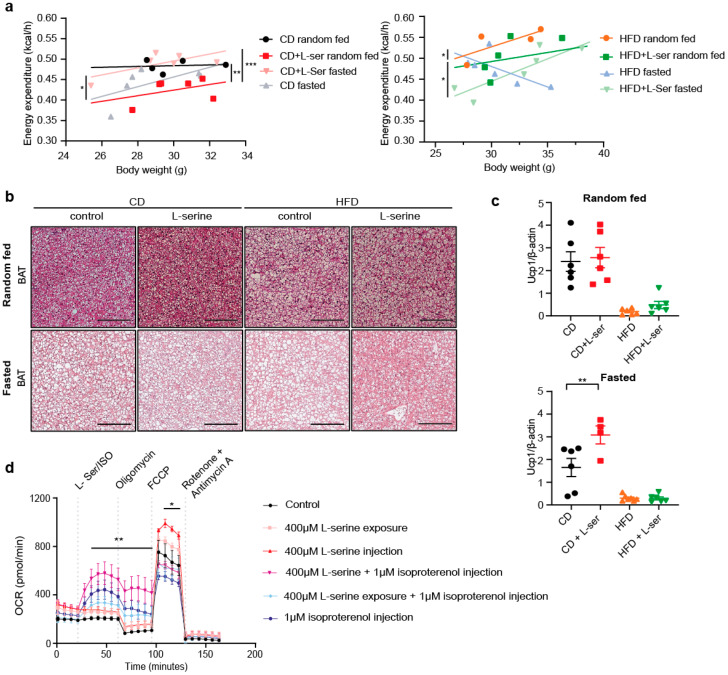
L-serine supplementation alters energy expenditure and potentiates mitochondrial respiration. (**a**) Energy expenditure after overnight fasting in relation to body weight of random fed and fasted CD (chow diet) and HFD (high-fat diet)-fed mice with or without L-serine supplementation at weeks 7 (CD) or 8 (HFD) (CD random fed: *n* = 5; CD fasted, CD+L-ser random fed, CD+L-ser fasted: *n* = 6; HFD random fed: *n* = 4; HFD fasted, HFD+L-ser random fed, HFD+L-ser fasted: (*n* = 6). (**b**) Representative H&E stainings of brown adipose tissue (BAT) from random fed and fasted CD-fed and HFD-fed mice with or without L-serine supplementation after 8 weeks starting at 4 (random fed) and 8 (fasted) weeks of age. Scale bars show 100 µM. (**c**) Quantification of uncoupled protein 1 (UCP1) protein content normalized to beta actin from BAT of random fed and fasted CD-fed and HFD-fed mice with or without L-serine supplementation (*n* = 6). (**d**) Oxygen consumption rate (OCR) of differentiated primary brown adipocytes pre-treated or not with 400 µM L-serine and acutely exposed (injected) to 400 µM L-serine and/or 1 µM isoproterenol (*n* = 3 replicates from 2 mice). Statistical analysis in panel a is described in Section 2.10. Data are shown as mean ± SEM, one or two-way ANOVA with Tukey’s post-hoc test. * *p* < 0.05, ** *p* < 0.01,*** *p* < 0.001.

## Data Availability

All primary data can be obtained from the corresponding author upon reasonable request.

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
