# Peer review of "L-Serine Supplementation Blunts Fasting-Induced Weight Regain by Increasing Brown Fat Thermogenesis"

_nutrients, 2022, doi:10.3390/nu14091922_

Round 1

Reviewer 1 Report

The effects of L-serine supplementation to the diet, especially in high-fat diets, are shown in this paper. The authors conclude that L-serine supplementation blunted fasting induced body weight regain, especially in diet-induced obese mice. The methodological approach has been carried out by chronic treatment with L-Ser (from the beginning, for 8 or 16 weeks). It would have been interesting to know the results if L-Ser had been added after fasting and starting from an already established overweight, to observe whether weight gain occurs or not. Although these results are interesting, it would have been added after having developed obesity, which is the most frequent situation (as a potential therapy).

Specific comments.

-There are aspects in the description of the figures that sometimes are not clear. For example in Figure 1 and also in figure 2 (figure legends), it is indicated that they are results obtained at week 8 of treatment with CD or HFD (with or without L-ser). Are both figures at 8 weeks?  Then the results are contradictory. There must be an error. I understand that Figure 2 shows the results at 16 weeks of treatment and Figure 1 at 8 weeks, but it is not clear (see page 5, line 195: supplemented with or without L-serine for eight weeks (n=8)) and page 7 line 220: CD (chow diet) or HFD (high-fat diet) with or without L-serine supplementation for eight weeks (n=6)).

-Related to result 3.1. L-serine supplementation does not alter body weight or glucose homeostasis in ad libitum fed mice:

  • It should be indicated in the statement of the result in which week this result is given (8 weeks) since subsequently this is not the case.
  • The authors say: "L-serine supplementation did not alter body weight (Fig. 1a) or body composition (Fig. 1b)." In graph 1a it seems to be differences in the HFD and HFD+L-ser curves from 6 weeks onwards, or at least a change in trend. The curves begin to separate at 6 weeks. Since the graph only shows the initial 8 weeks of HFD treatment and in methods it is stated that they went up to 16 weeks, it becomes necessary to continue the curve up to 16 weeks to show that there really was no difference between them. Or else it cannot be stated that L-ser did not alter body weight or it should be stated that this is so up to week "x".
  • It would also be advisable to quantify the area under the curve in the TTGs to make sure that there is no difference between HFD and HFD+L. Ser (as stated in Figure 1e). Apparently, the peak and curve of HFD is below the HFD+L-Ser and the same in the case of  CD (although to a lesser extent).
  • In the histological image in FIGURE 1 (F): the size of adipocytes in f (PGF) treated with HFD seems to be (in the image shown) larger in size than those fed control diet (CD). When the images are quantified (Supplementary Figure 1a) it is observed that there are only 3 mice and that the results are not significant. There are very high values (in one mouse) and the other two are similars to CD. It is logical that the results can be very variable between mice (being mice fed ad libitum), but then the “n” is too low to say that there are no differences in the sizes of adipocytes of HFD with respect to CD. And certainly, the image is not representative of the statement made (pag4 line 181-182: “H&E stainings of the liver, subcutaneous (SCF) and perigonadal (PGF) fat did not indicate any signs of inflammation (Fig. 1f) or changes in adipocyte cell size (Fig. S1a))”. It is surprising that adipocyte size in PGF does not increase with HFD. Has the size been assessed in cell surface or cell radius values?

- Supplementary Figure 2e shows a loss of day/night rhythm in CD fasted and CD specially in CD+L-Ser fasted. Any ideas as to why. Some comment or discussion would be advisable.

MINOR COMMENTS

- Pag 6 line 216: figure 2f is the figure 2e

- Pag 10 line 306: However, Uncoupled protein

- Homogenize the time format: e.g.: pag 2 line 78: at week eight or 16. pag 3 line three hours. Review all manuscript

- Page 5 of Supplementary: figures are cropped at bottom

Author Response

Reviewer 1

Comments and Suggestions for Authors

The effects of L-serine supplementation to the diet, especially in high-fat diets, are shown in this paper. The authors conclude that L-serine supplementation blunted fasting induced body weight regain, especially in diet-induced obese mice. The methodological approach has been carried out by chronic treatment with L-Ser (from the beginning, for 8 or 16 weeks). It would have been interesting to know the results if L-Ser had been added after fasting and starting from an already established overweight, to observe whether weight gain occurs or not. Although these results are interesting, it would have been added after having developed obesity, which is the most frequent situation (as a potential therapy).

We would like to thank the reviewer for taking the time to provide her/his expert opinion on our study. We agree with the reviewer that a number of additional experiments will be of great interest to further investigate the role of L-serine supplementation in regulation of body weight. We consider the integration of an L-serine arm in one of the many ongoing clinical weight loss studies as most likely the most informative future experiment. However, we are currently still working in the details and approvals of such a study, which is why we cannot provide these data at the moment. Furthermore, the proposed intervention studies in mice are certainly also of interest. However, we believe that this would be beyond the scope of the current manuscript. As these experiments are not covered by our animal protocol we would need several months (currently ~6 months) for governmental approval and then additional 16-20 weeks for the experiment.

Specific comments.

  • There are aspects in the description of the figures that sometimes are not clear. For example in Figure 1 and also in figure 2 (figure legends), it is indicated that they are results obtained at week 8 of treatment with CD or HFD (with or without L-ser). Are both figures at 8 weeks? Then the results are contradictory. There must be an error. I understand that Figure 2 shows the results at 16 weeks of treatment and Figure 1 at 8 weeks, but it is not clear (see page 5, line 195: supplemented with or without L-serine for eight weeks (n=8)) and page 7 line 220: CD (chow diet) or HFD (high-fat diet) with or without L-serine supplementation for eight weeks (n=6)).

We would like to apologize if we have not made the differences between the experimental setup in Figures 1 and 2 clear. Figure 1 describes the metabolic response in ad libitum fed mice to chronic L-serine supplementation. Conversely, Figure 2 addresses the impact of repeated overnight fasting as indicated in Figure 2a on various metabolic parameters.

Both figures report data from mice after 8 weeks on the respective diets. However, the difference is that mice in Figure 1 were fed ad libitum, with no major impact of L-serine supplementation, whereas there is decreased body weight (re-)gain in L-serine supplemented mice after fasting. In addition, we started the experiments in Figure 1 with 4 week old mice, whereas the data in Figures 2-4 were started at an age of 8 weeks. We now also included body weight data as Supplementary Figure 1a for mice fed either CD or HFD ad libitum with or without L-serine supplementation for 16 weeks starting at an age of 8 weeks. We hope that this answer could adequately address the reviewers concern.

  • Related to result 3.1. L-serine supplementation does not alter body weight or glucose homeostasis in ad libitum fed mice:

It should be indicated in the statement of the result in which week this result is given (8 weeks) since subsequently this is not the case.

We hope that our explanation above could clarify the reviewers concern. In addition we now also include body weight data in ad libitum fed mice for up to 16 weeks.

  • The authors say: "L-serine supplementation did not alter body weight (Fig. 1a) or body composition (Fig. 1b)." In graph 1a it seems to be differences in the HFD and HFD+L-ser curves from 6 weeks onwards, or at least a change in trend. The curves begin to separate at 6 weeks. Since the graph only shows the initial 8 weeks of HFD treatment and in methods it is stated that they went up to 16 weeks, it becomes necessary to continue the curve up to 16 weeks to show that there really was no difference between them. Or else it cannot be stated that L-ser did not alter body weight or it should be stated that this is so up to week "x".

We did not detect any statistically significant differences in the body weight of ad libitum fed mice upon L-serine supplementation during our 8 week study. Between weeks 7 and 8 of the cohort shown in Figure 1 we performed tolerance tests, which resulted in weight loss of some of the mice that was not recovered until the end of the experiments. However, as stated above this difference was not statistically significant. We could have excluded those mice from Figure 1a, but thought due to data consistency we will report data from all mice. To corroborate our finding we now show body weight data of another cohort that was monitored for 16 weeks as Supplementary Figure 1a, where we also do not observe a difference in body weight.

  • It would also be advisable to quantify the area under the curve in the TTGs to make sure that there is no difference between HFD and HFD+L. Ser (as stated in Figure 1e). Apparently, the peak and curve of HFD is below the HFD+L-Ser and the same in the case of CD (although to a lesser extent).

We now include baseline subtracted AUC values for the GTT data in Figure 1e, which confirmed the absence of a statistically significant difference.

  • In the histological image in FIGURE 1 (F): the size of adipocytes in f (PGF) treated with HFD seems to be (in the image shown) larger in size than those fed control diet (CD). When the images are quantified (Supplementary Figure 1a) it is observed that there are only 3 mice and that the results are not significant. There are very high values (in one mouse) and the other two are similars to CD. It is logical that the results can be very variable between mice (being mice fed ad libitum), but then the “n” is too low to say that there are no differences in the sizes of adipocytes of HFD with respect to CD.

We now changed the data presentation of Supplementary Figure 1c to show the size distribution of adipocytes within the different depots and treatment groups. We measured >10.000 cells per mouse of three mice per group. These data now show that HFD increases the proportion of very large adipocytes (>2400 mm2) at the expense of smaller ones. However, we did not observe a difference between control and L-serine supplemented mice.

  • And certainly, the image is not representative of the statement made (pag4 line 181-182: “H&E stainings of the liver, subcutaneous (SCF) and perigonadal (PGF) fat did not indicate any signs of inflammation (Fig. 1f) or changes in adipocyte cell size (Fig. S1a))”. It is surprising that adipocyte size in PGF does not increase with HFD. Has the size been assessed in cell surface or cell radius values?

We now changed the presentation of the data in Supplementary Figure 1b to better show the overall size distribution of the adipocytes. We measured surface area. These data show that HFD feeding increases the relative proportion of large adipocytes at the expense of smaller adipocytes. However, we did not observe differences between control and L-serine supplemented mice within each diet group. We now also include qPCR data on inflammatory markers in SCF and PGF as Supplementary Figure 1b.

  • Supplementary Figure 2e shows a loss of day/night rhythm in CD fasted and CD specially in CD+L-Ser fasted. Any ideas as to why. Some comment or discussion would be advisable.

We would like to thank the reviewer for raising this interesting point. The data indicate that mice after fasting also feed during the light cycle and that this is more pronounced in L-serine supplemented mice, which is also seen in the food intake data (Fig. S3B). We explain this observation in lines 283-284.

 MINOR COMMENTS

  • Pag 6 line 216: figure 2f is the figure 2e

We corrected this mistake

  • Pag 10 line 306: However, Uncoupled protein

We corrected the error

  • Homogenize the time format: e.g.: pag 2 line 78: at week eight or 16. pag 3 line three hours. Review all manuscript

We changed the format according to the reviewer suggestion

  • Page 5 of Supplementary: figures are cropped at bottom

This must have been a printing issue on the side of the reviewer as our PDF is complete.

Reviewer 2 Report

In this work, the authors investigated the effect of L-Serine supplement on metabolism in a control diet versus high fat diet male mouse model. In addition, the authors investigated the effect of fasting/refeeding on weight regain. L-serine is an amino acid that plays an important role in the production of proteins, DNA, and cell membranes. It is a nonessential amino acid, however, because of the diverse roles that serine plays in the body, dietary intake may offer certain benefits. L-serine has been shown to positively correlate to insulin secretion and sensitivity and L-serine metabolism is altered in Type 2 diabetes. The manuscript aim is of interest but a lot of results remain to be clarified.

Major comments:

  1. It is somewhat surprising that male C57Bl6 mice fed a HFD for 8 weeks do not show significant increased body weight, fat mass and glucose and insulin levels. Please explain.
  2. Moreover, there is a discrepancy between BW at 8-weeks and the increased fat and lean mass compared to controls.
  3. Please explain how serine supplementation tend to increase glucose level.
  4. It is stated in the legend of figure 1 that glucose level was measured in fed state. What about insulin?
  5. The authors claim that there is no inflammation in tissues, but no inflammatory markers have been stained. The H&E data cannot support this conclusion.
  6. The authors claim no changes in adipocyte size with the HFD while the pictures (and the literature) show increase adipocyte size during HFD feeding in mice compared to CD.
  7. Similarly, the absence of hepatic lipid droplets is surprising in male, the Suppl. Figure 1 showed even a tend to decrease. Please explain.
  8. How do you explain that fat mass is 2 to 3 times higher in Figure 2 than in Figure 1? What was the body weight of these mice?
  9. In Fig.2 PGF + SCF is about 1g for CD and 2 g for HFD while TF is between 3 g and 10 g. Where those differences come from?
  10. Fig.2 Do the authors have the liver weight/TG content and H&E-stained liver as in figure 1
  11. Fig.2 same as figure 1. Add the magnification
  12. What about the female mice?
  13. Line 302-304, the authors claim that lipid droplets size was reduced, based on the H&E staining. I think this is a lot of extrapolations. Lipid content of the whole BAT by Folch extraction would be needed to confirm the statement.
  14. A large body of literature shows that UCP1 expression is induced by HFD as compared to CD, this is called metabolic thermogenesis and will prevent the development of obesity during HFD feeding. https://doi.org/10.1152/ajpregu.00411.2010

The magnitude of UCP1 increase is pretty variable within the literature, however, it is surprising that the authors find a net reduction in the current study.

  1. The authors show that liver weight during HFD + L-ser was lower as compared to HFD alone (Fig.S1d) together with lower hepatic glycogen content and no changes in PEPCK and PPT. This is somewhat difficult to understand. The authors should further develop their
  2. L369-370 Activation of BAT do not induced EE. BAT activation produces heat so the author statement in not correct and should be rewritten.
  3. Measurement of insulin level during the OGTT would have help to better define the potential role of L-serine in glucose homeostasis and insulin sensitivity. Especially that the OGTT and PTT were unchanged.
  4. As it is now, the authors should tone down the “possible mechanism” by which L-serine will control fat regain and energy expenditure. No real mechanism has been demonstrated in the experiment, it is mainly speculation.

Minor comments:

  1. Please add the scale for H&E pictures
  2. There is no figure 2F, please correct
  3. Typo error in Fig.3c “dark phase”

Author Response

Reviewer 2

In this work, the authors investigated the effect of L-Serine supplement on metabolism in a control diet versus high fat diet male mouse model. In addition, the authors investigated the effect of fasting/refeeding on weight regain. L-serine is an amino acid that plays an important role in the production of proteins, DNA, and cell membranes. It is a nonessential amino acid, however, because of the diverse roles that serine plays in the body, dietary intake may offer certain benefits. L-serine has been shown to positively correlate to insulin secretion and sensitivity and L-serine metabolism is altered in Type 2 diabetes. The manuscript aim is of interest but a lot of results remain to be clarified.

Major comments:

  1. It is somewhat surprising that male C57Bl6 mice fed a HFD for 8 weeks do not show significant increased body weight, fat mass and glucose and insulin levels. Please explain.

We would like to thank the reviewer for raising this important point as we discovered an error in reporting the age of the mice used in Figure 1. In fact the mice were 4 weeks old at the beginning of the study. Thus, albeit the HFD mice gained more weight and the fat mass was significantly increased the young age most likely explains the lack of a stronger increased in body weight or impaired glucose tolerance. We now also include body weight data from mice that we followed for 16 weeks, starting at an age of 8 weeks.

  1. Moreover, there is a discrepancy between BW at 8-weeks and the increased fat and lean mass compared to controls.

The total body weight, fat mass and lean mass were not significantly different between L-serine treated and control mice within one diet group. However, HFD fed mice gained more body fat compared to CD fed mice.

  1. Please explain how serine supplementation tend to increase glucose level.

We did not observe any statistically significant increase in blood glucose levels upon L-serine supplementation.

  1. It is stated in the legend of figure 1 that glucose level was measured in fed state. What about insulin?

Both glucose and insulin were measured in random fed animals as stated in the figure legend.

  1. The authors claim that there is no inflammation in tissues, but no inflammatory markers have been stained. The H&E data cannot support this conclusion.

We now include data on gene expression of inflammatory markers as Supplementary Figure 1b as well as in Supplementary Figure 2e.

  1. The authors claim no changes in adipocyte size with the HFD while the pictures (and the literature) show increase adipocyte size during HFD feeding in mice compared to CD.

We now provide a more detailed analysis of the adipocyte size distribution as Supplementary Figure 1c. These data show that HFD increases the relative abundance of large adipocytes at the expense of smaller adipocytes. However, we did not observe differences between control and L-serine supplemented mice within each diet group.

  1. Similarly, the absence of hepatic lipid droplets is surprising in male, the Suppl. Figure 1 showed even a tend to decrease. Please explain.

We agree with the reviewer that this result was unexpected, but is in line with the body weight and glucose tolerance data. We conclude that this is due to the young age of the mice. Please see our answer to your point number 8.

  1. How do you explain that fat mass is 2 to 3 times higher in Figure 2 than in Figure 1? What was the body weight of these mice?

We would like to sincerely apologize for not properly indicating the difference in starting age between Figure 1 and Figure 2-4. The experiments shown in Figure 1 were performed in mice between the age of 4-12 weeks, whereas the subsequent experiments investigated mice starting at 8 weeks of age. We have now included body weight data for ad libitum CD and HFD fed control and L-serine supplemented mice as Supplementary Figure 1a. This difference in age explains the difference in fat mass between Figure 1 and Figure 2.

  1. In Fig.2 PGF + SCF is about 1g for CD and 2 g for HFD while TF is between 3 g and 10 g. Where those differences come from?

Inguinal subcutaneous adipose tissue and perigonadal adipose tissue are amongst the largest adipose depots in mice. However, there are multiple additional fat depots throughout the body. Moreover, the EchoMRI quantifies all fat mass, including ectopic fat in skeletal muscle and liver.

  1. Fig.2 Do the authors have the liver weight/TG content and H&E-stained liver as in figure 1

We now include data on liver weight and liver TG as well as histology of the liver in Supplementary Figure 2 a-c.

  1. Fig.2 same as figure 1. Add the magnification

All histological images include a scale bar.

  1. What about the female mice?

This is a very interesting and of course important aspect. Unfortunately, we did not study female mice in this project. The main reason for this is that female C57/Bl-6 mice are largely protected from diet induced obesity and insulin resistance. Thus, we limited our study to male mice.

  1. Line 302-304, the authors claim that lipid droplets size was reduced, based on the H&E staining. I think this is a lot of extrapolations. Lipid content of the whole BAT by Folch extraction would be needed to confirm the statement.

We agree with the reviewer that quantification of total lipid content would provide additional data. However, as we refer to lipid droplet size and not total lipid content we are convinced that our statement is correct as stated in the manuscript.

  1. A large body of literature shows that UCP1 expression is induced by HFD as compared to CD, this is called metabolic thermogenesis and will prevent the development of obesity during HFD feeding. https://doi.org/10.1152/ajpregu.00411.2010. The magnitude of UCP1 increase is pretty variable within the literature, however, it is surprising that the authors find a net reduction in the current study.

We agree with the reviewer that there are conflicting data in the literature as discussed by Fromme and Klingenspor in the cited publication. We can only speculate that differences in duration of high fat diet feeding, age of mice, genetic background and housing temperature can contribute to this. To this end, we can only report our findings as shown in Figure 4

  1. The authors show that liver weight during HFD + L-ser was lower as compared to HFD alone (Fig.S1d) together with lower hepatic glycogen content and no changes in PEPCK and PPT. This is somewhat difficult to understand. The authors should further develop their

We agree with the reviewer. We performed a number of additional experiments to further investigate the role of L-serine in hepatic metabolism. However, we were unable to find an explanation for the observed differences.

  1. L369-370 Activation of BAT do not induced EE. BAT activation produces heat so the author statement in not correct and should be rewritten.

We fully agree with the reviewer but do not understand the reviewers comment. We state that activation of BAT increases EE and not the opposite.

  1. Measurement of insulin level during the OGTT would have help to better define the potential role of L-serine in glucose homeostasis and insulin sensitivity. Especially that the OGTT and PTT were unchanged.

We agree with the reviewer. Unfortunately, our animal protocol did not allow repeated blood draw to measure insulin secretion, which is why we cannot provide these data.

  1. As it is now, the authors should tone down the “possible mechanism” by which L-serine will control fat regain and energy expenditure. No real mechanism has been demonstrated in the experiment, it is mainly speculation.

We were surprised by the reviewers’ comment, as we were very careful in not overstating the potential mechanisms underlying the observed phenotype. We explicitly state that the molecular mechanisms remain to be determine. We now also state that activation of BAT is a “potential mechanism” not excluding additional factors.

Minor comments:

  1. Please add the scale for H&E pictures

All our histology pictures contain scale bars

  1. There is no figure 2F, please correct

We apologize for this mistake and have corrected the manuscript accordingly.

  1. Typo error in Fig.3c “dark phase”

We apologize for this mistake and have corrected the manuscript accordingly.

Reviewer 3 Report

The authors conducted a very interesting reserche on the L-serine supplementation role in fasting induced weight regain  The experiments were carefully prepared and the measurements are well described so it is easy to understad whole presented work. The results obtained are novel and reliable. Desplite it I have some sugestions for the authors :

Introduction

Some more information on the role of L-serine in the regulation of body weight, should be metioned due the fact that it is the main part of the articel . Authors only metioned that it possess various functions but it is not enouht to undertake such complex research.

Methods

2.9. Statistical Analysis. - why it is writen in italics???

Results

Figure 1. L - Diagrams A, B, C showing Body weight, Body composition and Lean mass are presented in a way that makes it difficult to observe changes and the presented trends, their size and quality is insufficient.

Disscustion

What are the aspects of the weight ragaint and systematic inflamation process - mayby role of L-serine supplementation should be metioned. Thera are some data that increased weight is strongly corealated with systematic inflamation and metabolic disorders. 

Various other amino acids are also known to play a role during fasting, (arginine, leucine, isoleucine, threonine), why authors did not showe their concentractions in relation to L-serine. Mayby it could play a role in reduction of body weight and fat mass?

Authrs should disscuss with other works that examine the brown fat thermogenesis - in order to compare the effectiveness of the presented method.

Author Response

Reviewer 3

Comments and Suggestions for Authors

The authors conducted a very interesting reserche on the L-serine supplementation role in fasting induced weight regain  The experiments were carefully prepared and the measurements are well described so it is easy to understad whole presented work. The results obtained are novel and reliable. Desplite it I have some sugestions for the authors :

Introduction

Some more information on the role of L-serine in the regulation of body weight, should be metioned due the fact that it is the main part of the articel . Authors only metioned that it possess various functions but it is not enouht to undertake such complex research.

 As stated in our introduction; L-serine plays an important role in different diseases and physiological functions. However, there are no detailed studies on the role of L-serine in the regulation of body weight. We no cite and refer to the study by Zhou et al demonstrating that long term treatment with L-serine reduces food intake and thereby body weight.

Methods

2.9. Statistical Analysis. - why it is writen in italics???

We apologize for this formatting error and have corrected this.

Results

Figure 1. L - Diagrams A, B, C showing Body weight, Body composition and Lean mass are presented in a way that makes it difficult to observe changes and the presented trends, their size and quality is insufficient.

We do not understand the reviewers concern as both line graphs and dot plots are standard ways of presenting the data.

Disscustion

What are the aspects of the weight ragaint and systematic inflamation process - mayby role of L-serine supplementation should be metioned. Thera are some data that increased weight is strongly corealated with systematic inflamation and metabolic disorders.

We would like to thank the reviewer for bringing up this important point. We now included qPCR data on adipose tissue inflammation that did not, beyond the expected reduction upon reduced body fat mass, show any direct effects of L-serine supplementation on this parameter. We agree with the reviewer that L-serine could modulate immune cell function. However, we did not specifically test this aspect in our current study and therefore would like to refrain from detailed discussion on this point.

Various other amino acids are also known to play a role during fasting, (arginine, leucine, isoleucine, threonine), why authors did not showe their concentractions in relation to L-serine. Mayby it could play a role in reduction of body weight and fat mass?

We agree with the reviewer. Unfortunately, we did not assess serum amino acid levels in our study. We included this limitation in our discussion.

Authrs should disscuss with other works that examine the brown fat thermogenesis - in order to compare the effectiveness of the presented method.

We are confident that the role of BAT in regulation of systemic metabolism is well known to the readers of Nutrients. Therefore, we did not include a detailed discussion on the role of BAT in regulation of systemic metabolism as this would exceed the scope of our manuscript. Furthermore, much of the potential discussion would be very speculative, which, as indicated by the reviewers above, should be omitted.

Round 2

Reviewer 1 Report

The manuscript has improved.